# Dataset of Psychological Scales and Physiological Signals Collected for Anxiety Assessment Using a Portable Device

Mohamed Elgendi [1,2,*] , Valeria Galli [2] , Chakaveh Ahmadizadeh [2] and Carlo Menon [1,2]

1    MENRVA Research Group, School of Mechatronic Systems Engineering, Simon Fraser University, Burnaby, BC V5A 1S6, Canada
2    Biomedical and Mobile Health Technology Lab, ETH Zurich, 8008 Zurich, Switzerland
\*    Correspondence: moe.elgendi@hest.ethz.ch

**Abstract:** Portable and wearable devices are becoming increasingly common in our daily lives. In this study, we examined the impact of anxiety-inducing videos on biosignals, particularly electrocardiogram (ECG) and respiration (RES) signals, that were collected using a portable device. Two psychological scales (Beck Anxiety Inventory and Hamilton Anxiety Rating Scale) were used to assess overall anxiety before induction. The data were collected at Simon Fraser University from participants aged 18–56, all of whom were healthy at the time. The ECG and RES signals were collected simultaneously while participants continuously watched video clips that stimulated anxiety-inducing (negative experience) and non-anxiety-inducing events (positive experience). The ECG and RES signals were recorded simultaneously at 500 Hz. The final dataset consisted of psychological scores and physiological signals from 19 participants (14 males and 5 females) who watched eight video clips. This dataset can be used to explore the instantaneous relationship between ECG and RES waveforms and anxiety-inducing video clips to uncover and evaluate the latent characteristic information contained in these biosignals.

**Keywords:** detecting anxiety; biosignal-based anxiety assessment; anxiety screening; improving anxiety management; individual anxiety level prediction; digital health

## 1. Background & Summary

Wearable devices (WDs) and artificial intelligence (AI) have become increasingly prevalent in our daily lives. WDs are also being used in healthcare settings to manage health conditions via easy-to-use mobile applications, AI-enabled conversational chatbots, and remote monitoring platforms. To date, advancements in WDs and AI have mostly overlooked how humans respond to stressful daily life situations. Thus, there is a need for an AI-based WD solution that can detect instantaneous stress and recommend possible actions to take in response.

Everyone experiences situations and events that trigger anxiety, and everyone reacts differently. High levels of anxiety can have a negative effect on one's quality of life. Anxiety scales are currently used to diagnose anxiety disorders; however, they are not practical for everyday use as a form has to be filled out, and the users have to score themselves. Note that these scales only reflect the user's feelings when filling them out. Convenient, easy-to-use WDs that collect physiological signals and monitor instantaneous anxiety levels continuously are thus urgently required.

According to a recent study, electrocardiogram (ECG) and respiration (RES) signals are the physiological signals most closely associated to stress [1]. However, not all stress

and anxiety studies have included ECG and RES signals [2–14]. Moreover, most researchers who collected physiological signals for stress and anxiety assessments have not made their datasets publicly available [2,4–6,8,10–18]. In experimental environments, video stimuli are often used to induce anxiety and stress, but several researchers have not used such stimuli [2–16,18–22]. Additionally, some researchers only used negative stimuli [2,6,23,24], making it difficult to distinguish between positive and negative stress induction. Interestingly, some researchers did not use any stimuli [2–16,18–22], making it even more difficult to assess instantaneous stress induction.

Thus, the purpose of this study was to investigate the feasibility of using physiological signals (i.e., ECG and RES signals) collected with a portable device to assess instantaneous anxiety and monitor its transition from a positive to a negative event, and from a negative to a positive event.

## 2. Methods

### 2.1. Participants

Nineteen volunteers (14 males and 5 females; ages: $26.15 \pm 8.69$ years; age range: 18–56 years) from different cultural backgrounds participated in the data collection experiment. These participants were students and staff from Simon Fraser University, Canada. An overview of the data in terms of gender is shown in Figure 1.

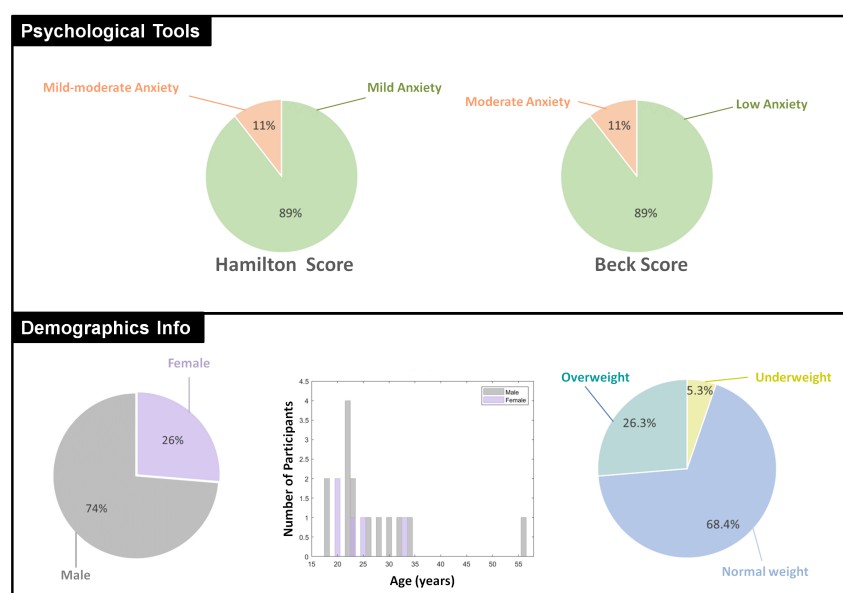

**Figure 1.** Overview of the psychological scales and demographic information. Most participants were young males who were assessed with low anxiety based on their psychological scales result.

We confirmed that all participants were proficient in English, understood the experiment description, and completed a consent form. General information about the study was verbally provided after the participants registered for the experiment. When the participants agreed to register for the study, detailed instructions for the experiment were read out to them, and a physical copy of the consent form was handed to them, clearly stating the following: "If you volunteer as a participant in this study, you will be asked to fill out two questionnaires and watch different media (movie clips, audio, texts, etc.) that can cause certain levels of anxiety, fear, sadness, or happiness while two non-invasive sensors are placed on your hands/arms and chest. Certain clips contain materials that may be disturbing to certain viewers. The session will take up to an hour, and you will be seated during the experiment. The final decision about participation is yours, and you can withdraw at any time".

### 2.2. Ethics Statement

Our study was reviewed and approved by the Simon Fraser University Research and Ethics Committee (2019s0041).

### 2.3. Experimental Design

The experiment was set up to induce anxiety in an alternating manner: participants were first shown a negative video (anxiety induction), followed by a positive video (non-anxiety induction). To examine the transitional effects, the videos in each viewing session were arranged sequentially so that the resulting video sequence was the same for all participants. The first six-second video (later called Clip 0) was used to help the participants relax and prepare for the experiment.

### 2.4. Experimental Protocol

We described the experiment both orally and in writing before initiating the procedure. After answering all the questions raised by the participants, they were asked to sign an informed consent form. Thereafter, a brief introduction to the experiment was provided, and the participants were asked to take off their jacket and one sock (right leg). Thereafter, ECG and RES sensors were attached to the participants, who then sat in front of a 24″ flat LCD screen. The seat was about 1 m away from the table with the screen. The participants were asked if they had been diagnosed with any health condition, and all said they were healthy. The sensors were properly placed and examined before data collection; if both sensors functioned properly, the experiment began. During the experiment, the first author (ME) sat 1.5 m away from the participant to ensure that the experiment ran smoothly and that the data were collected simultaneously. All the participants completed the experiment at different times. The sensors were removed at the end, and the participants were encouraged to provide verbal feedback and suggestions. The study protocol is shown in Figure 2, and all participants followed it.

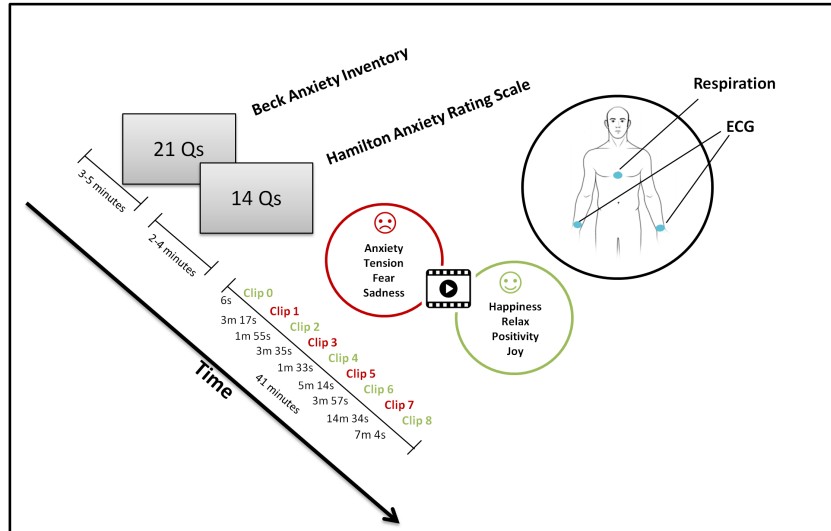

**Figure 2.** Study protocol. Each participant completed the full protocol, starting with the Beck Anxiety Inventory, then the Hamilton Anxiety Rating Scale, and then watched video clips with and without anxiety induction. The electrocardiogram and respiration signals were collected while watching the video clips using a portable device.

### 2.5. Videos

2.5.1. Quality-Check Session

The quality-check session aimed to examine the quality of the RES and ECG signals without explicitly training participants on the different types of anxiety-inducing (negative experience) and non-anxiety-inducing (positive experience) videos.

- **Clip 0.** A quality-check session: A short series of pictures. The clip duration was 6 s.

Since the data acquired during the quality-check session were not recorded, they were not included in the final dataset. This step was needed to record ECG and RES signals to ensure successful collection before starting the experiment.

2.5.2. Experiment Videos

The main goal of this study was to examine the effects of videos that induced anxiety versus those that did not. The experiment featured two standardized questionnaire-based tests: the Beck Anxiety Inventory with 21 questions (duration: 3–5 min) and the Hamilton Anxiety Rating Scale with 14 questions (duration: 2–4 min), as well as a 41 min video, as shown in Figure 2. Table 1 shows all the participants' corresponding anxiety assessments using both anxiety scales. An overview of the data in terms of the anxiety scales is presented in Figure 1.

The video consisted of eight clips (Clips 1–8) preceded by a short preparatory clip (Clip 0):

- **Clip 1**. A negative experience (anxiety-inducing): *The Present*, a 2014 animated short film that won 59 international awards. The short film explores the challenging topic of disability and living with an amputated leg. It had 7,822,858 views on YouTube (23 November 2021). https://www.youtube.com/watch?v=3XA0bB79oGc. The duration of the clip is 3 min and 16 s.
- **Clip 2**. A positive experience (non-anxiety-inducing): A cheerful children's choir singing about how music brings happiness. The original video clip has been removed from YouTube. It had 77 views on YouTube (23 November 2021). https://www.youtube.com/watch?v=Z_CB7IjjggY. The clip duration is 1 min and 59 s.
- **Clip 3**. A negative experience (anxiety-inducing): Anya, a short film that earned several award nominations over the span of 20 years. The film explores the sad life of a Russian orphan. It had 678,967 views on YouTube (23 November 2021). https://www.youtube.com/watch?v=RdHyCwPvppI. The clip duration is 3 min and 38 s.
- **Clip 4**. A positive experience (non-anxiety-inducing): A series of photographs depicting happy or funny moments with short captions and cheerful music in the background. It had 3,502,628 views on YouTube (23 November 2021). https://www.youtube.com/watch?v=JxJsai5nkGI. The clip duration is 1 min and 37 s.
- **Clip 5**. A negative experience (stress-inducing): A long series of car accidents filmed live. It had 1,548,780 views on YouTube (23 November 2021). https://www.youtube.com/watch?v=TkidANiymRw. The clip duration is 5 min and 17 s.
- **Clip 6**. A positive experience (non-anxiety-inducing): A video featuring the characters of the animated cartoon "Minions" dancing on the notes of Pharrell William's hit "Happy". It had 92,496,747 views on YouTube (23 November 2021). https://www.youtube.com/watch?v=MOWDb2TBYDg. The clip duration is 3 min and 51 s.
- **Clip 7**. A negative experience (anxiety-inducing): A series of natural disasters with documentary-style explanations. It had 6,905,750 views on YouTube (23 November 2021). https://www.youtube.com/watch?v=8bBKENJHZYc. The clip duration is 14 min and 42 s.
- **Clip 8**. "The world's most relaxing film" is a short video filmed along the west coast of Zealand in Denmark and released by a Danish tourism association in 2015. It shows beautiful and peaceful natural scenarios. The original video clip has been removed from YouTube. It had 8171 views on YouTube (23 November 2021). https://www.youtube.com/watch?v=dkFNdABPhC0. The clip duration is 7 min and 0 s.

**Table 1.** Participant classification based on psychological scales. Participant A06 was assessed using the Beck Anxiety Inventory and the Hamilton Anxiety Rating Scale to have low anxiety. Two subjects (A01 and A016) were assessed to have low anxiety with Hamilton Anxiety Rating Scale and Beck Anxiety Inventory, respectively.

| Participant | A01 | A02 | A03 | A04 | A05 | A06 | A07 | A08 | A09 | A10 | A11 | A13 | A14 | A15 | A16 | A18 | A19 | A20 | A21 |
|---|---|---|---|---|---|---|---|---|---|---|---|---|---|---|---|---|---|---|---|
| Beck Score | 20 | 11 | 0 | 9 | 13 | 25 | 10 | 2 | 7 | 2 | 14 | 12 | 1 | 7 | 23 | 6 | 7 | 4 | 7 |
| Beck (Low Anxiety) | ✓ | ✓ | ✓ | ✓ | ✓ | | ✓ | ✓ | ✓ | ✓ | ✓ | ✓ | ✓ | ✓ | | ✓ | ✓ | ✓ | ✓ |
| Beck (Moderate Anxiety) | | | | | | ✓ | | | | | | | | | ✓ | | | | |
| Hamilton Score | 20 | 12 | 5 | 4 | 12 | 18 | 12 | 1 | 6 | 4 | 11 | 14 | 4 | 6 | 15 | 13 | 9 | 3 | 3 |
| Hamilton (Mild Anxiety) | | ✓ | ✓ | ✓ | ✓ | | ✓ | ✓ | ✓ | ✓ | ✓ | ✓ | ✓ | ✓ | ✓ | ✓ | ✓ | ✓ | ✓ |
| Hamilton (Mild–Moderate Anxiety) | ✓ | | | | | ✓ | | | | | | | | | | | | | |

The anxiety-inducing videos were identified by searching for the following terms: "Sad videos that will make you cry after watching", "Heart-stopping & scary road moments caught on camera", and "top 10 videos that will make you sad". In contrast, the non-anxiety-inducing videos were "funniest video" and "top 10 videos that will make you happy". The reason behind the phrase choice as opposed to others, such as "anxious feeling video", is to ensure finding video clips that are well validated by a large population. The videos were pre-verified based on the YouTube ratings and comments confirming the elicited emotions (i.e., either positive or negative) based on a large sample size. For this reason, positive- and negative-emotions-inducing clips did not necessarily have the same duration. Moreover, the clips' durations were not restricted, to provide a diverse representation of anxiety- and non-anxiety-inducing scenarios, representing real-life events where the stimulus duration would not be fixed. The first clip featured a short film exploring the life of an individual coping with a daily struggle, whereas the last one featured a relaxing natural scenario that served as a "cool-down" phase before the end of the experiment. It is worth noting that the participants were introduced to these videos for the first time, and all participants confirmed that they had not seen the videos before the experiment.

*2.6. Sensors and Instruments*

The sensors used for the experiment were selected based on our previous work [1]. The MP45 portable data acquisition unit with two certified human-safe input channels and built-in amplifiers from Biopac Systems Inc. (Goleta, CA, USA) was used. The system is accompanied by the BSL 4 interactive software, electrode lead sets, headphones, disposable electrodes, abrasive pads, a USB cable, and a power transformer. The specifications of the MP45 system can be found here: https://www.biopac.com/wp-content/uploads/MP36-MP45.pdf (accessed on 23 November 2021).

2.6.1. ECG Sensor

An ECG sensor was recorded the electrical activity of cardiac muscles during contraction. ECG data were acquired at the wrists and right leg (triangular configuration) using a two-channel fully shielded cable assembly (SS2LB). First, the skin placement site was prepared by cleaning it with an alcohol (70% isopropanol) pad. The ECG electrodes were then pre-gelled and placed on the cleaned sites. This sensor also was preamplified and filtered the detected electric signal. The specifications of the SS2LB ECG module can be found here: https://www.biopac.com/product/lead-set-shielded-bsl/ (accessed on 23 November 2021).

2.6.2. Respiration Sensor

RES signals are generated when the chest expands and contracts. A Biopac TSD201 belt was placed on the chest around the same level as the heart. It is worth noting that the RES belt was placed above the participants' clothes to simulate daily life activities, rather than directly on the skin. A strain gauge transducer was embedded in the Biopac RES belt to quantify the magnitude of chest contractions and expansions. The specifications of the TSD201 belt can be found here: https://www.biopac.com/product/respiratory-effort-trans-tp/ (accessed on 23 November 2021).

**3. Data Records**

The dataset is publicly available in the Figshare repository https://figshare.com/articles/dataset/Anxiety_Dataset_2022/19875217 (accessed on 25 May 2022).

*Metadata*

The metadata are publicly available via this link: https://github.com/Elgendi/Anxiety-Dataset-MetaData-2022 (accessed on 25 May 2022). We present auxiliary information about the experiments in three metadata Excel files. The first Excel file, named

"Demographic Data", contains the participants' number, sex, age, height, and weight. The second Excel file, named "Anxiety Scales", contains the participants' responses to the Hamilton Anxiety Scale and Beck Anxiety Inventory.

## 4. Technical Validation

### 4.1. Qualitative Validation

We developed a validated protocol and stimuli to induce anxiety based on well-established research in this field [25,26]. Before starting the experiment, M.E. visually inspected the quality of the ECG and RES signals. M.E. began the experiment and saved the data once the ECG features were streamed in high quality and the RES signal was responding to the participant's inhalation and exhalation. Figure 3 shows the changes in the amplitudes of the ECG and RES signals from their baselines.

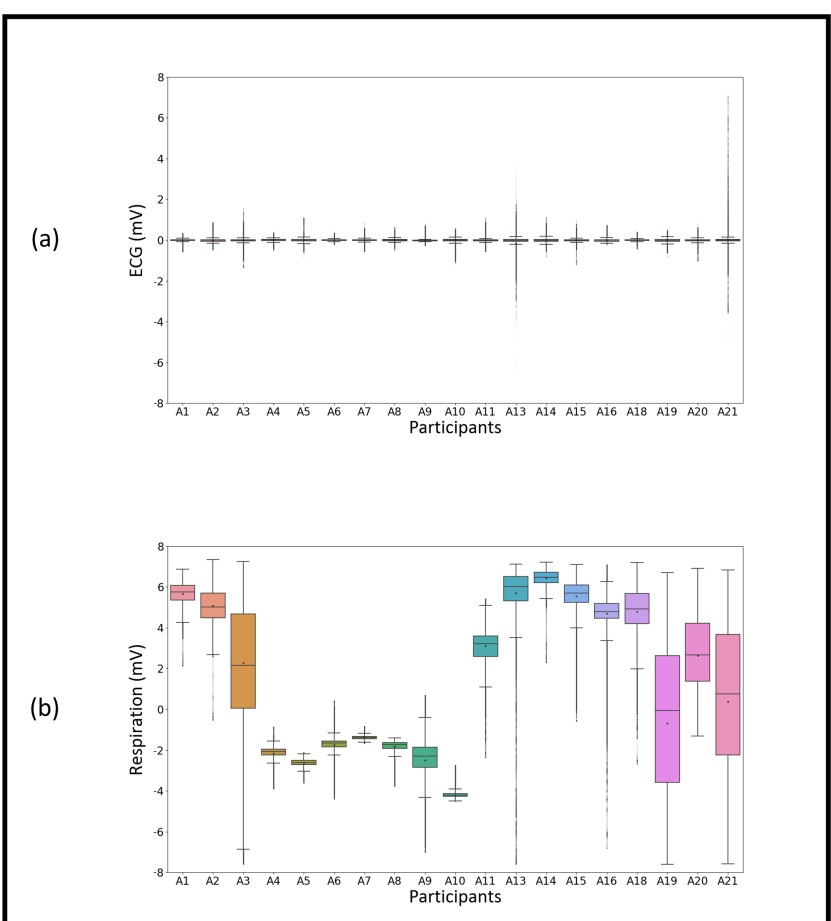

**Figure 3.** Boxplot of the (**a**) ECG and (**b**) RES signals collected for each participant. This figure shows the overall amplitude behaviour of the ECG and RES signals. The ECG signals are clearly more consistent (amplitude does not fluctuate) than the RES signals (amplitude fluctuates) for all the participants. Note that ECG = electrocardiogram signal and RES = respiration signal.

Note that the amplitude deviation (fluctuation) in the ECG signals was less than that of the RES signals. In other words, ECG was more stable in amplitude measurement than RES was. Table 2 shows a comparison of all the signals collected from all participants. Figure 4 provides a visual example of anxiety induction in real time. The morphological changes in participants who scored low and high on the anxiety scales can also be seen in Figure 4.

**Table 2.** Statistical description of the ECG and RES signals for all the participants. Note that ECG = electrocardiogram signal and RES = respiration signal.

| Participant | | A01 | A02 | A03 | A04 | A05 | A06 | A07 | A08 | A09 | A10 | A11 | A13 | A14 | A15 | A16 | A18 | A19 | A20 | A21 |
|---|---|---|---|---|---|---|---|---|---|---|---|---|---|---|---|---|---|---|---|---|
| ECG | Mean [mV] | 0.00 | 0.00 | 0.00 | 0.00 | 0.00 | 0.00 | 0.00 | 0.00 | 0.00 | 0.00 | 0.00 | −0.15 | 0.00 | 0.00 | 0.00 | 0.00 | 0.00 | 0.00 | 0.01 |
| | Std [mV] | 0.08 | 0.11 | 0.10 | 0.07 | 0.09 | 0.06 | 0.10 | 0.06 | 0.09 | 0.11 | 0.10 | 0.46 | 0.12 | 0.06 | 0.10 | 0.05 | 0.10 | 0.09 | 0.20 |
| | Median [mV] | −0.01 | −0.03 | −0.02 | −0.01 | −0.02 | −0.01 | −0.01 | −0.01 | −0.02 | 0.00 | −0.02 | −0.02 | −0.03 | −0.02 | −0.03 | 0.00 | −0.02 | −0.02 | −0.01 |
| | Range [mV] | 0.94 | 1.41 | 2.91 | 0.93 | 1.94 | 0.64 | 1.49 | 1.23 | 1.11 | 1.75 | 1.70 | 9.87 | 1.95 | 2.20 | 1.01 | 0.88 | 1.40 | 1.83 | 12.00 |
| | SNR [dB] | 49.46 | 51.29 | 46.23 | 49.39 | 51.96 | 49.41 | 53.17 | 47.23 | 53.26 | 52.52 | 42.18 | 26.98 | 37.82 | 31.59 | 41.53 | 36.80 | 34.90 | 45.72 | 44.62 |
| RES | Mean [mV] | 5.66 | 5.08 | 2.26 | −2.16 | −2.63 | −1.74 | −1.36 | −1.81 | −2.49 | −4.20 | 3.10 | 5.71 | 6.44 | 5.55 | 4.69 | 4.80 | −0.69 | 2.64 | 0.38 |
| | Std [mV] | 0.54 | 0.86 | 2.56 | 0.33 | 0.16 | 0.32 | 0.14 | 0.27 | 0.88 | 0.12 | 0.92 | 1.67 | 0.37 | 0.84 | 0.93 | 1.12 | 3.82 | 1.77 | 3.92 |
| | Median [mV] | 5.76 | 5.02 | 2.16 | −2.07 | −2.61 | −1.66 | −1.39 | −1.73 | −2.30 | −4.21 | 3.21 | 6.02 | 6.48 | 5.71 | 4.80 | 4.92 | −0.06 | 2.66 | 0.75 |
| | Range [mV] | 4.76 | 7.87 | 14.86 | 3.06 | 1.53 | 4.84 | 0.86 | 2.38 | 7.71 | 1.78 | 7.80 | 14.71 | 4.91 | 7.69 | 13.93 | 9.89 | 14.29 | 8.22 | 14.43 |
| | SNR [dB] | −20.80 | −18.74 | −1.04 | −24.50 | −26.97 | −19.29 | −27.20 | −21.38 | −12.43 | −31.24 | −21.96 | −16.96 | −25.77 | −19.57 | −16.28 | −14.79 | 5.98 | −4.68 | 4.92 |

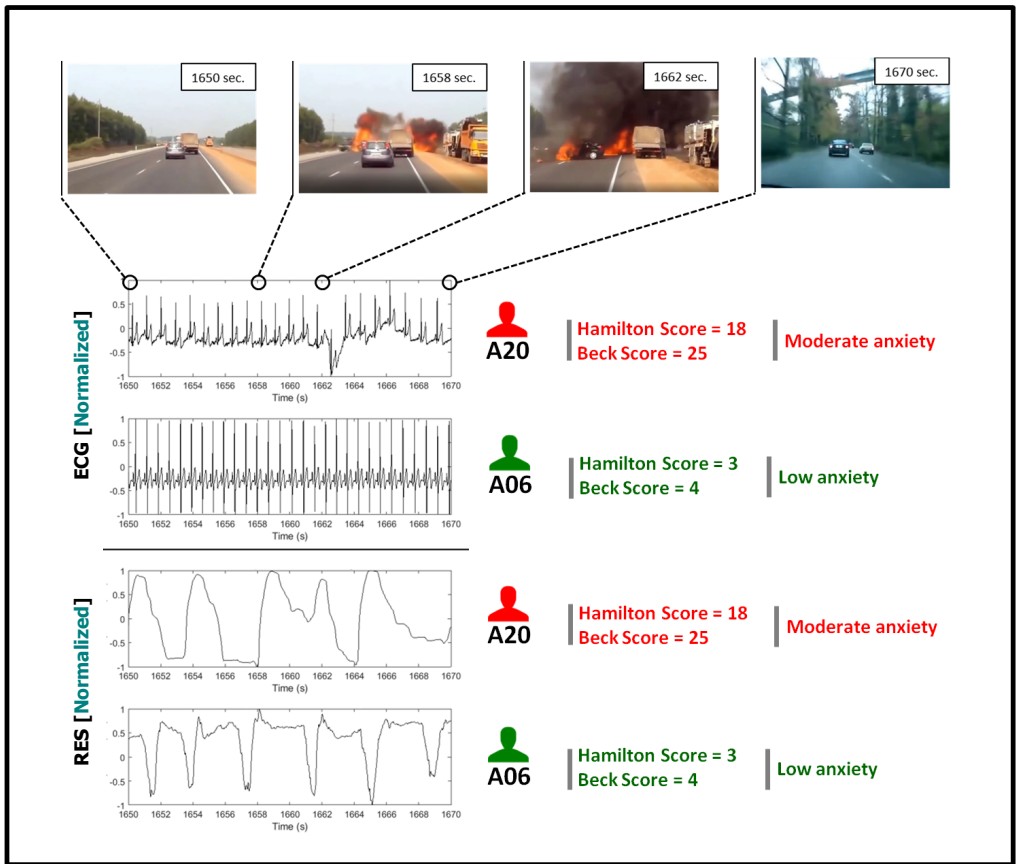

**Figure 4.** An example showing electrocardiogram (ECG) and respiration (RES) signals in participants with and without anxiety induction. Participant A06 was assessed using the Hamilton Anxiety Rating Scale to have low anxiety, while participant A20 was assessed to have moderate anxiety. Interestingly, the ECG showed a sudden change for participant A20 at the 1662nd second as a response to the anxiety-inducing event (car accident), while the ECG of participant A06 did not change. The same holds for the respiration signal. Note that the sudden change in RES morphology was visible before the change in ECG signal occurred.

*4.2. Quantitative Validation*

The quality of each signal was evaluated using a signal-to-noise ratio (SNR). We created two SNR algorithms to calculate the SNR of diverse ECG and RES signals. A third-order Savitzky–Golay filter with a frame length of 7 was used to create the first SNR algorithm for the ECG signals [27]. A third-order 0.05–1 Hz bandpass Butterworth filter was used to create the second SNR algorithm for the RES signals [28]. All the SNR scripts used are available in the project's GitHub repository (https://github.com/Elgendi/Anxiety-Dataset-MetaData-2022, accessed on 25 May 2022). We calculated the SNR for the ECG and RES signals, as shown in Table 2. The mean SNR of the ECG signals ranged from 26.98 dB to 53.26 dB, while that of the RES signals ranged from −31.29 dB to 4.99 dB. The calculated SNR indicated that the ECG signals were of a higher quality than the RES signals.

*4.3. Previous Studies*

Bosse et al. demonstrated that video materials can effectively induce both positive and negative emotional states [29]. Some studies have used video clips that displayed either happy, positive, or relaxing scenes to induce a positive emotional state or video clips that displayed scary, anxious, and sad scenes to stimulate a negative emotional state [30]. Previous studies highlighted the importance of alternating induced emotions by ensuring that each consecutive clip corresponds to a different target emotion [31,32]. In this study, eight video clips were placed together in an alternating fashion (positive/negative) and

preceded by a short introductory clip (Clip 0) for about 41 min (Figure 2). In prior studies, the physiological parameters typically examined were RES rates, ECG-based heart rates, and ECG-based heart rate variability. The correlation between (or changes in) RES rate and anxiety state has long been recognized and was previously investigated in multiple studies [33,34]. Masaoka and Homma extensively investigated the correlation between RES-related parameters and diverse forms of anxiety, such as the increase in RES rate in response to trait anxiety [35]. Conversely, several studies have considered ECGs and variations in heart rate as significant features for anxiety detection [36]. In particular, ECG-related features and heart rate variability have been used as features in machine learning classification models for anxiety detection, with accuracies ranging from 69% using the R–R interval [37] to 96% when combining multiple heart rate variability features collected from a wearable ECG sensor [38].

## 5. Usage Notes

The dataset is publicly available and can be downloaded via this link: https://figshare.com/articles/dataset/Anxiety_Dataset_2022/19875217 (accessed on 25 May 2022). The data were saved in .mat format, while the metadata were saved in .xls format. The dataset can be used to test hypotheses on anxiety and non-anxiety induction, develop an algorithm for detecting anxiety during real-time anxiety induction, and differentiate between transitional anxiety and non-transitional anxiety. The data were collected using a portable device and should attract the attention of scientists with different backgrounds, such as psychologists, health data scientists, and biomedical engineers who are interested in developing wearable solutions for anxiety detection.

*Limitations and Future Work*

The duration of exposure is relevant because the induced psychological state does not suddenly stop after ceasing the stimulus (i.e., concluding the video), which may influence the effect on the response to a subsequent stimulus [39]. Nonetheless, there is no recognized protocol for determining the appropriate duration of audio–visual stimuli. Some studies reported using the same duration for positive, negative, and neutral clips [29,31], whereas others used the same range but not necessarily an equal duration for the clips [30,40]. In particular, studies on physiological responses to anxiety used various durations of neutral stimuli, ranging from a few seconds of a neutral stimulus in between other emotions [40] to the same duration of positive and negative stimuli [29,31]. One important point to consider in terms of stimulus duration is that different types of stimuli elicit emotions of varying durations; for example, negative stimuli are known to impress higher arousal than positive ones when neutral videos are used as a baseline [39]. Another important point to consider when establishing the duration of stimuli is the excitation-transfer theory, which postulates that an individual will experience a state of residual arousal for a subsequent period after a stimulus without being aware of it; additional stimuli during this period may cause an emotional state that superimposes onto the previous one, making it difficult to distinguish between the effects of either stimulus. Note that the number of participants included in the study was limited, and the population was not balanced between male and female participants. Future work will need to consider a larger and gender-balanced sample and possibly a more diverse population. One of our next steps is to develop an algorithm to differentiate between anxiety-inducing and non-anxiety-inducing events. Moreover, we will examine if ECG is more sensitive than RES in detecting anxiety-inducing events.

## 6. Code Availability

The code can be accessed on the public GitHub link https://github.com/Elgendi/Anxiety-Dataset-MetaData-2022 (accessed on 25 May 2022).

**Author Contributions:** M.E. led the study and conducted the experiment. M.E., C.A., V.G. and C.M. conceived the study. All authors have read and agreed to the published version of the manuscript.



**Funding:** This research received no external funding.

**Institutional Review Board Statement:** The study protocol was approved by the Simon Fraser University Research and Ethics Committee (2019s0041).

**Informed Consent Statement:** Informed consent was obtained from all subjects involved in the study.

**Data Availability Statement:** https://figshare.com/articles/dataset/Anxiety_Dataset_2022/19875217 (accessed on 25 May 2022).

**Acknowledgments:** The authors would like to thank Kim Lajoie (Simon Fraser University, Canada) for her help with participant recruitment.

**Conflicts of Interest:** The authors declare no conflict of interest.

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
