# Peer review of "Dataset of Psychological Scales and Physiological Signals Collected for Anxiety Assessment Using a Portable Device"

_data, 2022_

Round 1

Reviewer 1 Report

The authors have designed the study and collected the dataset that would be valuable for future research and development. The study design is appropriate. Below are some of my comments:

1. The sample size is small. It is highly recommended to increase the sample size, that would be beneficial for any future studies utilizing this dataset.

2. Table 2 need to be revised. It seems that the authors are comparing the amplitudes but it is not clear in the table. Adding units would be helpful. Also, 'Respiration Rate' mentioned in the table doesn't seem to be appropriate; shouldn't it be Respiration? Please clarify.  

Author Response

Reviewer #1 (Remarks to the Author):

The authors have designed the study and collected the dataset that would be valuable for future research and development. The study design is appropriate. Below are some of my comments

Author response
: We thank the reviewer for the positive feedback.

Author action: None.

  1. The sample size is small. It is highly recommended to increase the sample size, that would be beneficial for any future studies utilizing this dataset.

Author response: We thank the reviewer for the positive feedback.

Author action: We acknowledge that the sample size utilized in the present study is small and added a sentence in the Limitation section addressing this point (lines 243-245):

“Note the number of participants included in the study was limited, and the population was not balanced between male and female participants. Future work will need to consider a larger and gender-balanced sample and possibly a more diverse population.”

  1. Table 2 need to be revised. It seems that the authors are comparing the amplitudes, but it is not clear in the table. Adding units would be helpful. Also, 'Respiration Rate' mentioned in the table doesn't seem appropriate; shouldn't it be Respiration? Please clarify. 

Author response: We thank the reviewer for the feedback; indeed, “respiration rate” is incorrect.

Author action 1: Table 2 has been modified with the correct label that is indeed “RES” as respiration. Units have been added, specifically “mV” for the mean, standard deviation, range, and median values and “dB” for signal-to-noise ratio (SNR).

Reviewer 2 Report

This paper describes a dataset that can be used to examine the impact of anxiety-inducing videos on biosignals; i.e., electrocardiogram (ECG) and respiration (RES) signals. 

The background and purpose are described well. In the methods section, it is not clear how specifically the participants were recruited, what was the distribution of staff versus students, and why there was an uneven distribution of male and female participants.

The experimental design is not clearly described, I think that is mainly due to the sentence "All participants watched different video stimuli for the experiment." I initially interpreted this sentence as stating that all videos were different for all participants, i.e., that there were 19 different videos, a different one for each individual participant. From figure 2, that is given much later in the paper, I understand that the positive an negative clips were alternated within a video that was exactly the same for all participants, which makes it a time-series design, not a randomized between-subjects design. The clips used differ in length, with the positive experience clip being on average much shorter (most of them all less than 2 minutes) than the negative experience clips (all are longer than 3 minutes and one is even 14 minutes). This difference is acknowledged and explained in the limitations, but could also be mentioned in the methods section explaining why this was decided.

Author Response

Reviewer #2 (Remarks to the Author):

This paper describes a dataset that can be used to examine the impact of anxiety-inducing videos on biosignals; i.e., electrocardiogram (ECG) and respiration (RES) signals. 

The background and purpose are described well. In the methods section, it is not clear how specifically the participants were recruited, what was the distribution of staff versus students, and why there was an uneven distribution of male and female participants.

Author response: We thank the reviewer for the positive feedback. The study was carried out at the university and a diverse population of student and one member of the stuff.  

Author action: We acknowledge that participant distribution was not gender balanced and addressed this in the Limitation section (lines 243-245):

“Note the number of participants included in the study was limited, and the population was not balanced between male and female participants. Future work will need to consider a larger and gender-balanced sample and possibly a more diverse population.”

The experimental design is not clearly described, I think that is mainly due to the sentence "All participants watched different video stimuli for the experiment." I initially interpreted this sentence as stating that all videos were different for all participants, i.e., that there were 19 different videos, a different one for each individual participant. From figure 2, that is given much later in the paper, I understand that the positive an negative clips were alternated within a video that was exactly the same for all participants, which makes it a time-series design, not a randomized between-subjects design.

Author response: We thank the reviewer for bringing up this lack of clarity.

Author action: We removed the sentence for clarity. Then we added in the Experimental protocol subsection the following sentence.

“The study protocol is shown in Figure2, and all participants followed it.”

The clips used differ in length, with the positive experience clip being on average much shorter (most of them all less than 2 minutes) than the negative experience clips (all are longer than 3 minutes and one is even 14 minutes). This difference is acknowledged and explained in the limitations, but could also be mentioned in the methods section explaining why this was decided.

Author response: We thank the reviewer for bringing up this point. The positive vs negative clips did not have the exact same duration because they were chosen based on user rating on YouTube as detailed in the Method section; moreover, the stimulus is usually investigated with a fixed duration but we sought to provide a more realistic approach.

Author action: We added a sentence in the Methods section clarifying the selection method for the clips and the duration disparity was added (lines 145-148):

“For this reason, positive and negative inducing emotions clips did not necessarily have the same duration. Moreover, the clips' duration was not restricted to provide a diverse representation of anxiety and non-anxiety inducing scenarios, representing real-life events where the stimulus duration would not be fixed.”

Reviewer 3 Report

Although the manuscript was well written, many issues force me to not recommend the current manuscript for publication.

Introduction:

Background literature was not thoroughly discussed. For example -- Lacks an explanation on how and why ECG and RES (including what specific variables such as HR or HRV) are associated with stress. No theory is mentioned. Indeed, a strong rationale for Author's research question is missing. What's the significance of the study? Why should people care? These are important question the introduction should address but are missing or are just poorly developed. Authors also provide no hypotheses or mention the exploratory nature of the study.  Lastly, strong statements are made without citations to support claims.

Methods:

Small sample. 

"All participants watched different video stimuli for the experiment" this statement gave me great pause as to how then you can make claims if they did not receive the same stimuli.

Was clip 0 baseline? 6 sec is not sufficient. Video clips lengths vary too much and is a potential confound. Moreover, Authors claim these videos were validated but provide no empirical evidence to prove this.

Provide psychometrics for measures used both reported and for the current study

Results & Discussion:

What data analyses were performed? Just descriptives? No data interpretations provided.

Author Response

Reviewer#3

Although the manuscript was well written, many issues force me to not recommend the current manuscript for publication.

Introduction:

Background literature was not thoroughly discussed. For example -- Lacks an explanation on how and why ECG and RES (including what specific variables such as HR or HRV) are associated with stress. No theory is mentioned. Indeed, a strong rationale for Author's research question is missing. What's the significance of the study? Why should people care? These are important question the introduction should address but are missing or are just poorly developed. Authors also provide no hypotheses or mention the exploratory nature of the study.  Lastly, strong statements are made without citations to support claims.

Author response: We thank the reviewer for bringing up this point. We tried to simulate real-life events where the stimulus would not be fixed. This paper was submitted to the journal called Data; therefore, it is a descriptive data paper; methodological articles on detecting anxiety using HRV on the same dataset will be published soon.

Author action: We added a sentence in the Methods section clarifying the selection method for the clips and the duration disparity was added (lines 145-148):

“For this reason, positive and negative inducing emotions clips did not necessarily have the same duration. Moreover, the clips' duration was not restricted to provide a diverse representation of anxiety and non-anxiety inducing scenarios, representing real-life events where the stimulus duration would not be fixed.”

Methods:

Small sample. 

Author response: We thank the reviewer for the positive feedback.

Author action: We acknowledge that the sample size utilized in the present study is small and added a sentence in the Limitation section addressing this point (lines 243-245):

“Note the number of participants included in the study was limited, and the population was not balanced between male and female participants. Future work will need to consider a larger and gender-balanced sample and possibly a more diverse population.”

"All participants watched different video stimuli for the experiment" this statement gave me great pause as to how then you can make claims if they did not receive the same stimuli.

Author response: We thank the reviewer for bringing up this lack of clarity.

Author action: We removed the sentence for clarity. Then we added in the Experimental protocol subsection the following sentence.

“The study protocol is shown in Figure2, and all participants followed it.”

Was clip 0 baseline? 6 sec is not sufficient. Video clips lengths vary too much and is a potential confound. Moreover, Authors claim these videos were validated but provide no empirical evidence to prove this.

Provide psychometrics for measures used both reported and for the current study

What data analyses were performed? Just descriptives? No data interpretations provided

Author response: We thank the reviewer for the valuable point. We addressed these points above. This is a data description paper submitted to the journal called Data; methodological papers showing the impact of anxiety will be published soon. A statistical analysis of the Data was provided. We did discuss the results and provided a limitation section.

Author action: None.

Round 2

Reviewer 3 Report

Although this is a data descriptive paper, it should be grounded in theory and provide a good justification for the data.  This, is still lacking in the paper. Although the figure is informative, Authors should also elaborate and clearly describe their data collection method in their method section. Although Authors claim their results were discussed, no significance is mentioned. Even though this is a data paper, what is the point to say that things differed or were of better "quality" if they as not statistically different from one another. This is misleading. The previous studies section is still lacking.

Author Response

Reviewer #3 (Remarks to the Author):

Although this is a data descriptive paper, it should be grounded in theory and provide a good justification for the data.  This is still lacking in the paper. Although the figure is informative, Authors should also elaborate and clearly describe their data collection method in their method section. Although Authors claim their results were discussed, no significance is mentioned. Even though this is a data paper, what is the point to say that things differed or were of better "quality" if they as not statistically different from one another. This is misleading. The previous studies section is still lacking.

Author response
: We thank the reviewer for the comments. We believe that the data collection method is thoroughly described, and no statistical significance was tested because the aim of the work is to provide a dataset for future analyses, thus we have not included statistical tests in this work.

Author action: We added the following the “Previous studies” section in the methods includes several examples of the available literature in the field.

Previous studies

“Bosse et al. demonstrated that video materials can effectively induce both positive and negative emotional states.[29] Some studies have used video clips that displayed either happy, positive, or relaxing scenes to induce a positive emotional state or video clips that displayed scary, anxious, and sad scenes to stimulate a negative emotional state.[30] Previous studies have highlighted the importance of alternating induced emotions by ensuring that each consecutive clip corresponds to a different target emotion.[31,32] In this study, eight video clips were put together in an alternating fashion (positive/negative) and preceded by a short introductory clip (Clip 0) for about 41~min (Figure2). In prior studies, the physiological parameters typically examined were RES rates, ECG-based heart rates, and ECG-based heart rate variability. The correlation between (or changes in) RES rate and anxiety state has long been recognized and was previously investigated in multiple studies.[33,34] Masaoka and Homma extensively investigated the correlation between RES-related parameters and diverse forms of anxiety, such as the increase in RES rate in response to trait anxiety.[35] Conversely, several studies have considered ECGs and variations in heart rate as significant features for anxiety detection.[36] In particular, ECG-related features and heart rate variability have been used as features in machine learning classification models for anxiety detection, with accuracies ranging from 69% using the R–R interval [37] to 96% when combining multiple heart rate variability features collected from a wearable ECG sensor.[38]”